# Exploring Quality of Life, Stress, and Risk Factors Associated with Irritable Bowel Syndrome for Female University Students in Taiwan

**DOI:** 10.3390/ijerph18083888

**Published:** 2021-04-07

**Authors:** Huan-Hwa Chen, Chich-Hsiu Hung, Ai-Wen Kao, Hsiu-Fen Hsieh

**Affiliations:** 1School of Nursing, Kaohsiung Medical University, No. 100, Shih-Chuan 1st Road, Kaohsiung 807, Taiwan; hwa@mail.hwai.edu.tw (H.-H.C.); hsiufen96@gmail.com (H.-F.H.); 2School of Nursing, Chung Hwa University of Medical Technology, No. 34, Wenhua 1st St., Tainan 717, Taiwan; 3Department of Medical Research, Kaohsiung Medical University Hospital, Kaohsiung Medical University, No. 100, Tzyou 1st Road, Kaohsiung 807, Taiwan; 4Department of Internal Medicine, National Cheng Kung University Hospital, No. 138, Shengli Road, Tainan 704, Taiwan; kaoaiwen@mail.ncku.edu.tw

**Keywords:** irritable bowel syndrome, prevalence, female university students, quality of life, stress

## Abstract

Irritable bowel syndrome (IBS) is a common recurrent functional gastrointestinal disorder that impacts on patients physically and mentally. Studies on IBS have focused on adults, yet few studies have examined IBS among female university students. The aim of this study was to investigate the prevalence of IBS for female university students and its related factors. Using a cross-sectional study design, a total of 2520 female university students were recruited in southern Taiwan. The structured questionnaires, including the Rome III IBS diagnostic questionnaire, IBS symptom severity scale, Perceived Stress Scale, and World Health Organization Quality of Life BREF questionnaire (WHOQOL-BREF) were used for data collection. A total of 1894 female students complete the questionnaires. The response rate was 75.15%. The results indicated 193 female students with IBS and the prevalence of IBS was 10.1%. IBS female students had higher levels of stress and lower QOL than non-IBS female students. The risk factors for female university students developing IBS were dysmenorrhea, food avoidance, class absenteeism, and the lower physical domain of QOL. It is advised to consider these factors when providing students with counselling and relevant services in the expectation of alleviating their IBS symptoms, reducing the incidence rate of IBS, and further improving their QOL.

## 1. Introduction

Irritable bowel syndrome (IBS) is a common recurrent functional gastrointestinal disorder. Its main symptoms are abdominal pain or discomfort accompanied by changes in bowel habits and stool appearance without abnormality in biochemical or tissue structures [1]. Approximately 1.1%–45% of world’s population have IBS [2]. The prevalence rate in Western countries is 5%–10% [2,3], and that in Southeast Asia is approximately 7%, with varying rates according to diagnostic criteria, and the geographical area will affect the reported IBS prevalence worldwide [2]. Although the prevalence rate in Southeast Asia is the lowest, the rate in Taiwan has reached 18%–22%, which is even higher than that in Western countries [4]. IBS is particularly prevalent among the population aged 20–50 years [5]. Its prevalence rate also differs between sexes. In general, the prevalence rate of women is approximately twice as high as that of men [2], yet in Taiwan, the rates of the two sexes are comparable [4].

Currently, the pathogenesis of IBS has not been clarified. However, most relevant studies have indicated that gastrointestinal function and symptoms are correlated with mental status. Studies have confirmed that a significant correlation exists between abnormal gastrointestinal movements and anxiety; patients with IBS demonstrate different degrees of disease experience under physical, mental, social, cultural, and environmental interactions [1]. IBS is also associated with stress, food, medication, and intestinal inflammation despite its unclear pathogenesis. Although IBS is not life-threatening, patients experience recurrent episodes, constantly seek medical treatment, and have to receive various clinical diagnoses, examinations, medication, and treatment, which substantially affect their daily routine, sleep quality, leisure activities, diet, and even employment, further leading to reduced work productivity, depression, and anxiety [6].

In fact, a study indicated that functional gastrointestinal symptoms are common in young adults regarded as a negative response to stress [7]. Also, patients with IBS are often reported to have depression, anxiety, or physical discomfort during medical consultation [8]. Moreover, their quality of life (QOL) is severely affected, but some studies revealed that poorer QOL is a risk factor of IBS [9,10]. Accordingly, IBS negatively influences patients physically and mentally. Many studies on IBS have focused on adults [4,11]. The IBS prevalence rate among university students in China ranges between 7.85% and 10.4% [12,13]. Those with IBS report higher degrees of anxiety and depression than did those without IBS [12]. Nevertheless, in Japan the prevalence of IBS was defined by various criteria ranges 19%–26%, respectively [14,15]. These above findings suggest that the prevalence of IBS is very high in young adults. 

Studies regarding the correlation between IBS and the menstrual cycle among Taiwanese female high-school students have been explored [16], yet information on the prevalence and the associated factors among female university students in Taiwan is relatively scant. Thus, there is a need to understand the relationship among the IBS prevalence, stress and QOL for female university students and its related factors.

## 2. Methods

### 2.1. Study Design

A cross-sectional study design was applied in this study.

### 2.2. Participants and Setting

This study was conducted in Chung Hwa University of Medical Technology from March to September 2016. Before collecting data, the principal investigator (PI) met the Deans of colleges of the university of medical technology to present this study. Female students aged 20 years or above at a university were recruited through convenience sampling. Those who had organic gastrointestinal disorders (e.g., tumor, ulcer, intestinal inflammation, and rectal bleeding), had received gastrointestinal surgeries, or could not independently fill out the questionnaire were excluded. The sample size was calculated based on G power 3.1.1 version in this study with the effect size of 0.15, a desired significance level of 0.05, two-tailed and a power of 0.95 [17]. The estimated minimal sample size was 580 participants. Assuming an attrition rate of 30%, a sample size of at least 754 was required. All fitted inclusion criteria 2520 female students were enrolled in the program. 

### 2.3. Data Collection and Ethecial Consideration

This research was approved by the research ethics committee of the university (REC-102-019-2) and after receiving approval from the Institutional Review Board at researchers’ university. The PI explained the purpose and processes of this study to the students during their class meetings and asked for their willingness to participate. After they signed an informed consent form, the PI obtained every student’s demographic characteristics, the Chinese version of the Rome III criteria for functional gastrointestinal disorder questionnaire, the severity of IBS questionnaire, the perceived stress scale, and the World Health Organization Quality of life-BREF Taiwan version. We recruited participants from 2520 female students aged 20 years or above and enrolled in the day program at a university of medical technology. After they signed an informed consent form, participants completed the questionnaires. A total of 1894 students completed the questionnaire with a response rate of 75.15%.

### 2.4. Instruments

A structured questionnaire was administered for data collection, including the demographic characteristics questionnaire, the Chinese version of the Rome III criteria for functional gastrointestinal disorder questionnaire, IBS symptom severity scale, the perceived stress scale, and the World Health Organization Quality of life-BREF Taiwan version.

#### 2.4.1. The Demographic Characteristics Questionnaire

The demographic characteristics questionnaire involves age, body height, body weight, colleges, years of education, living status, working status, relationship status, internship experience, current internship, exercise habits, dysmenorrhea, food avoidance, academic performance, class absenteeism, and medication (gastrointestinal medication and traditional Chinese medicine).

#### 2.4.2. The Chinese Version of the Rome III Criteria for Functional Gastrointestinal Disorder Questionnaire 

We adapted the Chinese version of the Rome III criteria for functional gastrointestinal disorder questionnaire [18]. The diagnostic criteria for IBS included recurrent abdominal pain or discomfort for at least 3 days/month during the last 3 months, with at least 6 months associated with two or more of the following features: (1) improvement with defecation; (2) symptoms associated with changes in defecation frequency; and (3) symptoms associated with changes in stool form. The specificity of the questionnaire was 87.8% and the test–retest reliability was 81.7% [19]. To ensure the questionnaire accorded with situations in Taiwan, we invited five researchers or experts in the fields of gastroenterology or gastroenterology nursing to evaluate the instrument. The content validity index was 0.90.

#### 2.4.3. The IBS Symptom Severity Scale (IBS-SSS)

The IBS-SSS was developed by Francis, Morris, and Whorwell [20] to evaluate the severity of IBS. The questionnaire is a type of visual analog scale and consists of five questions, each scoring from 0 (no pain) to 100 (worst pain) for a total score of 500. A score below 75 indicates a normal gastrointestinal condition; a score of 75–175, 175–300, and a score greater than 300 indicate mild, moderate, and severe IBS symptom, respectively [20]. The scale was separately translated by two bilingual experts (native Chinese speakers) from English to Chinese and back-translated by other two bilingual experts (native English speakers who had not seen the scale before) from Chinese to English. All experts examined the original English, translated Chinese, and back-translated English versions through group discussions and modified the questions of the Chinese version. Eventually, the Chinese version was employed for pretests with 30 patients with IBS from the outpatient department for final semantic revisions Cronbach’s α coefficient was 0.70.

#### 2.4.4. The Perceived Stress Scale (PSS)

We adopted the perceived stress scale developed by Cohen, Kamarck, and Mermelstein [21] and translated by Chu and Kao [22]. The scale is composed of 14 questions, with Questions 4, 5, 6, 7, 9, 10, and 13 being positively worded questions and the remainder being negatively worded. The scale was used to assess a patient’s perception of life stress in the recent month, particularly for measuring the degree of stress resulting from diseases and behavior disorders. Cronbach’s α coefficient was 0.85 [22]. A 5-point Likert scale is employed with each question scored from 0 to 4, corresponding to “never,” “rarely,” “sometimes,” “often,” and “always,” respectively. A higher score indicates a higher degree of perceived stress. The Cronbach’s α coefficient in this study was 0.73.

#### 2.4.5. The World Health Organization Quality of Life-BREF Questionnaire (WHOQOL-BREF)

The WHOQOL-BREF Taiwan version is a type of general health-related QOL measurement. It was simplified from the original 100 questions covering six domains into four domains, namely physiology (seven questions), psychology (six questions), social relations (four questions), and environment (nine questions) [23]. It consists of 26 questions with two additional indigenous questions due to cross-ethnicity (being respected and accepted) and culture (food), for a total of 28 questions. Each category was scored from 4 to 20 points, and the Cronbach’s α of the questionnaire was 0.91. The questions were scored using a 5-point Likert scale ranging from 1 to 5 points, indicating totally disagree, partly disagree, no comment, partly agree, and totally agree, respectively. Three of the questions in the scale are negatively worded. A higher total score suggests a more preferable QOL [24]. The Cronbach’s α coefficient in this study was 0.89. 

### 2.5. Statistical Analysis

We conducted statistical analyses using SPSS version 22.0 (IBM Corporation, Armonk, NY, USA). Descriptive statistical analyses were performed for the demographic characteristics. Inferential statistical analyses were conducted to explore the prevalence of IBS in terms of different colleges and compare the differences in demographic characteristics between students with or without IBS. The t test and chi-square analyses were conducted to examine the difference among all demographic variables between IBS and non-IBS groups. The multiple logistic regression was used to estimate the odds ratio (OR) with 95% confidence intervals (95% CI) for developing IBS among female university students.

## 3. Results

### 3.1. Participants’ Demographic Characteristics 

Among 2520 students, 1894 (75.15%) completed the questionnaire. The completed questionnaires, including 442 students from the College of Medical Technology (23.4%), 746 from the College of Nursing (39.4%), and 706 from the College of Human Life Science (37.2%; Table 1). The mean age of the students was 21.59 ± 1.40 years and their mean BMI was 21.29 ± 3.80 kg/m^2^. Table 2 indicates the demographic characteristics of the female university students.

### 3.2. Prevalence of IBS and Severity of IBS

Among the 1894 students, 193 had IBS, with a prevalence rate of 10.1%. Of the students with IBS, 96 were IBS-mix type (IBS-M) (50%), 45 were IBS-diarrhea type (IBS-D) (23.3%), 31 were IBS-constipation type (IBS-C) (16.1%), and 19 were IBS-unsubtyped (IBS-U) (9.8%) (Figure 1). The remaining two students with IBS, could not be classified by their IBS subtypes because they did not complete all required data. The Colleges of Medical Technology, Nursing, and Human Life Science had 52 (2.7%), 76 (4.0%), and 65 (3.4%) students with IBS, respectively (Table 1). Among the 193 students with IBS, 6.9%, 39.8%, 45.2%, and 8.1% had normal gastrointestinal conditions, mild IBS, moderate IBS, and severe IBS, respectively (Figure 2).

### 3.3. Demographic Characteristics of Female Students with or without Irritable Bowel Syndrome

The prevalence rates of the female university students with IBS were 11.76%, 10.18%, and 9.21% in the Colleges of Medical Technology, Nursing, and Human Life Science, respectively. No significant difference was found among the three colleges (*χ*^2^ = 1.943; *p* = 0.378). There were no significant differences in age, body height, body weight, BMI, colleges, years of education, living status, working status, relationship status, internship experience, current internship, and regular exercise habits. However, students with IBS experienced more dysmenorrhea than did those without IBS (52.6% vs. 25.7%, *χ*^2^ = 61.120, *p* < 0.001). The percentage of students avoiding certain types of food and having regular medication was significantly higher in those with IBS compared to non-IBS students (53.5% vs. 23.2%, *χ*^2^ = 79.818, *p* < 0.001 & 27.5% vs. 14.4%, *χ*^2^ = 22.360, *p* < 0.001). There was no significant difference between those with and without IBS regarding academic performance, but the percentage of those with class absenteeism for more than 3 days in semester was significantly higher in students with IBS than those without IBS (43.8% vs. 27.4%, *χ*^2^ = 22.259, *p* < 0.001) (Table 2). 

### 3.4. The Levels of Stress and QOL between Female Students with and without IBS

Students with IBS reported a significantly higher level of stress than did those without IBS (28.87 ± 6.80 vs. 26.06 ± 6.45, *t* = −5.66, *p* < 0.001) (Table 3). Moreover, those with IBS perceived having a significantly lower overall perception of QOL than did those without IBS (3.19 ± 0.64 vs. 3.42 ± 0.72, *t* = 4.73, *p* < 0.001) as well as a significantly lower overall perception of their health satisfaction (2.80 ± 0.73 vs. 3.27 ± 0.79, *t* = 8.33, *p* < 0.001). Students with IBS received significantly lower scores in the four domains compared with those without IBS (Table 3). 

### 3.5. The Risk Factors for IBS of Female Students

We conducted multiple logistic regression analyses to examine the IBS risk factors among the female university students. IBS was treated as the dependent variable, and dysmenorrhea, food avoidance, regular medication, class absenteeism, stress, and the four domains of QOL were the independent variables. The results revealed that dysmenorrhea, food avoidance, class absenteeism, and the physical domain of QOL were the influential factors of IBS. The risk of developing IBS for students with dysmenorrhea was 2.5 times higher than that of students without dysmenorrhea (95% CI: 1.804–3.526, *p* < 0.001). The risk for those who avoided particular foods was 3.2 times higher than that for those who did not (95% CI: 2.285–4.459, *p* < 0.001). The risk of those with absenteeism for more than 3 days /semester was 1.8 times higher than that for those without absenteeism for more than 3 days/semester (95% CI: 1.264–2.481, *p* = 0.001). The risk of those who received an additional point in the physical domain of quality of life was reduced by 13% (95% CI: 0.774–0.977, *p* = 0.019) (Table 4). 

## 4. Discussion

The prevalence of IBS among the female university students was 10.1%, which is within the range reported in China (8.7%–31.3%) [12,18,25], yet lower than the range in other countries (20.5%–41.8%) [26,27]. The variations reported in the rate of the prevalence of IBS might due to differing diagnostic criteria, lack of knowledge about various cross-cultural aspects of IBS, and geographical area. Among the 193 students with IBS, the IBS-M accounted for the most (50%), followed by IBS-D, IBS-C, and IBS-U. Most Chinese students were IBS-D [25] and IBS-M [18]. Study have indicated that microbial factors play main roles in IBS pathophysiology; IBS-D is related to small intestinal bacterial overgrowth and IBS-C is related to increased levels of methanogenic archaea [25].These results are similar to the results of studies in other Asian countries whose participants have a medical background. Previous study showed that most of the students who have a medical background tend to have IBS-M in Asia [28]. The reason may be related to the diversity of Asian diets, diet culture, and related taboos. Changes in diet affect the ecological environment of the intestinal tract, which in turn affects changes in intestinal symptoms in patients with IBS [29]. In addition, studies in humans have confirmed that the use of antibiotics play a significant role in the occurrence of IBS and changes in intestinal symptoms [30,31]. Therefore, the high proportion of antibiotics used clinically in Taiwan may also be a contributing risk factor to develop IBS [32].

In addition, we found that there was no significant difference in the prevalence of IBS among the three colleges. Students in the College of Medicine had a higher risk of developing functional bowel diseases compared with those in other colleges [2,26]. The lack of difference may be explained by the fact that all of the participants in this study were enrolled in medical-related departments despite being in different university colleges. Hence, similarities in the experiences of students in medical-related departments may result in little or no difference in the prevalence of IBS. 

In sum, our results indicated that dysmenorrhea, class absenteeism, food avoidance, and the physical domain of QOL were risk factors associated with IBS among female university students. The risk for students with dysmenorrhea was 2.5 times higher than those without dysmenorrhea. During their menstrual cycle, women may experience frequent and severe abdominal pains, abdominal distention, diarrhea, or constipation, which may be due to hormonal changes on visceral nerves [33]. Moreover, women with dysmenorrhea experienced more frequent IBS symptoms than did those without menstrual pain [34]. Relevant pathological studies on patients with IBS show that brain–intestine interactions, abnormal feelings in the viscera, abnormal bowel movement, and psychosocial factors are correlated with the incidence of IBS [35]. 

Additionally, we found that the female university students with absenteeism for more than 3 days had a 1.8-times higher risk of suffering from IBS. There are many possible reasons for students to be absent [36], such as stress, quality of teachers, social life, part time job, exams, and life events [37]. In Taiwan students who have physical health conditions, such as dysmenorrhea, IBS, and stomach flu may decide to take a day of menstrual or sick leave. In this study, it is possible that students with IBS missed classes because of the impact of IBS. Higher levels of absenteeism among students suffering from IBS may have cascading effects on their school performance. Our study also showed that the prevalence of IBS among the female university students with food avoidance was 3.2 times higher than those without food avoidance. In Taiwan, the diverse diet is a reflection of the cuisine and culture of China, Southeast Asia, Japan, and the West, incorporating various dietary taboos and cultures. Hence, young people likely to drink bubble tea and consume western fast food, such as pizza, fried chicken, and spicy food. This is one of the reasons why cultural diversity can make the intestines and stomach unwell. In this study, IBS students reported that they were more likely to avoid some of these foods. Previous studies have suggested that food was likely an essential cause of intestinal inflammation or intestinal symptoms [38], but the pathophysiological role of food in IBS is still vague and complicated. However, different types of eating habits have been proven to affect the diversity and function of the intestinal flora, which in turn triggers IBS or aggravates its symptoms. Certain foods, particularly those with fermentable oligosaccharides, disaccharides, monosaccharides, and polyols (FODMAPs), are reported to cause IBS symptoms [39]. In fact, approximately two-thirds of patients with IBS deemed food to be the cause of their symptoms [40], and approximately 90% implement food avoidance to prevent worsening symptoms [41]. Hence, dietary education plays a crucial role in reducing the incidence of IBS and the exacerbation of symptoms. However, patients should be aware of nutritional imbalances resulting from food avoidance.

In this study, we found that students take regular medication such as gastrointestinal medication and traditional Chinese medicine was not a risk predictor in IBS. It may be due to the convenience and accessibility of Taiwan’s health insurance [42] that these students can visit a doctor and take medication at any time if they feel unwell, without having to wait for a long time, thus resulting in regular medication risk not being a predictor. Stress has been demonstrated associated with the pathophysiology of functional gastrointestinal disorder and visceral hypersensitivity, as well as been found to be an exacerbating factor for gastrointestinal symptoms [43]. Stress activates hypothalamic–pituitary–adrenal (HPA) axis and results in the corticotrophin-releasing hormone being secreted form hypothalamus, subsequently altering the homeostasis of visceral sensitivity and the metabolic, immune, and autonomic nervous systems. Stress also has been shown to change the gut microbiota, permeability, and intestinal motility. Thus, stress alters the brain-gut-microbiota interaction, causing or exaggerating IBS [43]. 

Many studies also indicate that patients with IBS have significantly greater stress compared with those without IBS [14,44,45]. Those patients tend to develop anxiety due to abdominal discomfort and abnormal stool appearance, which could affect their long-distance travel, work, social interaction, and learning activities, and increase their perception of stress [45]. Therefore, IBS may influence students’ academic performance and future career development, and thus result in their overall higher levels of reported stress [46]. However, in our study, we found that stress was different from previous studies [14,44,45]. One possible reason could be that the stress of university students is different from that of adults. A previous study showed that the main stressors of university students were related to school performance, family, interpersonal relationships, future career development and self-expectations [47]. In addition, lots of stress factors related to family can affect university students’ learning, such as family socioeconomic status, family structure, and family resources. School-related stress factors include things, such as schoolwork, teacher-student interaction, whether to have a part-time job, the amount of credits for school learning, future career planning. Personal factors impacting stress include personality characteristics, previous learning achievements, gender, and dating. Studies have documented that all of the above factors are related to stress for learning [48]. In addition to many situations in the workplace, the stressors of adults could be associated with performance, competition, promotion, workload, interpersonal relationship with colleagues or bosses, and working environment. It may also be affected by other stressors such as family, relationships, and finances [49]. Therefore, the stress load of adults is much greater than that of university students. However, given that previous studies investigating the relationship between IBS and stress were mostly focused on adults [14,44,45] rather than on female university students, it may be one of the reasons why the results are different from the previous studies. Nevertheless, stress is still a risk factor in students that deserves attention. In the future, stress-related interventions can be made to prevent their stress from getting worse.

This study indicated that the scores in the four domains of QOL among the female university students with IBS were significantly lower than those without IBS. The quality of life of patients with IBS was significantly lower than that of patients with chronic diseases, such as hypertension, diabetes, and asthma, but comparable to that of patients with end-stage renal disease and depression [10,50]. Therefore, QOL is traditionally evaluated as the main outcome of various chronic diseases, but Siegrist’s study revealed that QOL is a risk factor of cardiovascular disease. After their study, some related studies have reported similar findings [51,52]. Studies have theorized that the possible mechanism is that the physical domain of QOL affects the autonomous nerve system (ANS), which in turn triggers or affects its prognosis [53]. However, ANS dysfunction is considered to be associated with many chronic diseases including IBS [54]. ANS dysfunction disrupts the brain-gut-microbiota axis function and communication between the central nervous system and enteric nervous system, subsequently changing the visceral sensitivity, gastrointestinal motility, secretion and immune functions [55]. Thus, ANS dysfunction may play a role in the pathogenesis of IBS. Whether poor QOL is a cause or effect of IBS is still being debated but it is indicated that poorer QOL is a risk factor of IBS [9]. However, Gralnek, Hays, Kilbourne, Naliboff, and Mayer considered it to be a result of IBS. In fact, poor QOL among patients with IBS may not be entirely caused by IBS [10]. One study demonstrated that half of the incidence of reduced QOL among patients with IBS resulted from other factors, such as anxiety [56]. Another study indicated that lower physical and mental QOL increased the probability of IBS [57]. Compared with that in men, the effect of physical QOL on the probability of IBS occurrence in women was more notable [57]. The results of this study confirmed that the physical domain of QOL among the female university students was a predictor of IBS. In other words, a bi-directional interaction existed between QOL and IBS. Therefore, alleviating the discomfort of IBS, identifying the factors influencing QOL and improving them would help reduce the incidence of IBS. Students with IBS had low self-perception of their overall QOL and were less satisfied with their personal health compared with those without IBS. The four QOL domains were significantly lower in those with IBS than in those without IBS. Therefore, IBS comprehensively influenced the QOL of the female university students.

IBS not only affected the female university students’ perception of stress but also their QOL. Many studies indicate similar results regardless of the instrument employed [58,59]. We found that female university students with IBS had greater stress and poorer QOL than did those without IBS. In Taiwan, most of the students studying in universities are likely to face pressures around their academic performance in technical courses as well as those associated with finances and interpersonal relationships. Thus, varying degrees of stress among students of technical and vocational schools may greatly affect their learning activities and general well-being. However, the stress derived from their absenteeism and poor academic performance were also risk factors for the incidence of IBS [58,60].

The limitation of the study is the use of a self-administered questionnaire which may result in a higher frequency of missed and inaccurate data compared to an interview-based study. IBS is a diagnosis of exclusion and an extensive diagnostic algorithm is necessary from the clinical test before we label the client as IBS. Therefore, some bias may appear in the students with IBS selected by the questionnaire. Also, we employed convenience sampling when recruiting female university students of medical technology in southern Taiwan. Therefore, the research may not generalize to all female university students in Taiwan. 

## 5. Conclusions

IBS is a chronic functional gastrointestinal disorder. The prevalence of IBS among female university students in this study was 10.1%, which is similar to that in Western countries. Dysmenorrhea, food avoidance, class absenteeism, and the physical health domain of QOL were identified as factors for IBS among female university students. It is advised to consider these factors when providing students with counseling and relevant services with the goal of alleviating their IBS symptoms, reducing the incidence rate of IBS, and further improving their QOL. Further research should include university students in northern Taiwan as well as male and female university students in order to better understand the role of gender in IBS.

## Figures and Tables

**Figure 1 ijerph-18-03888-f001:**
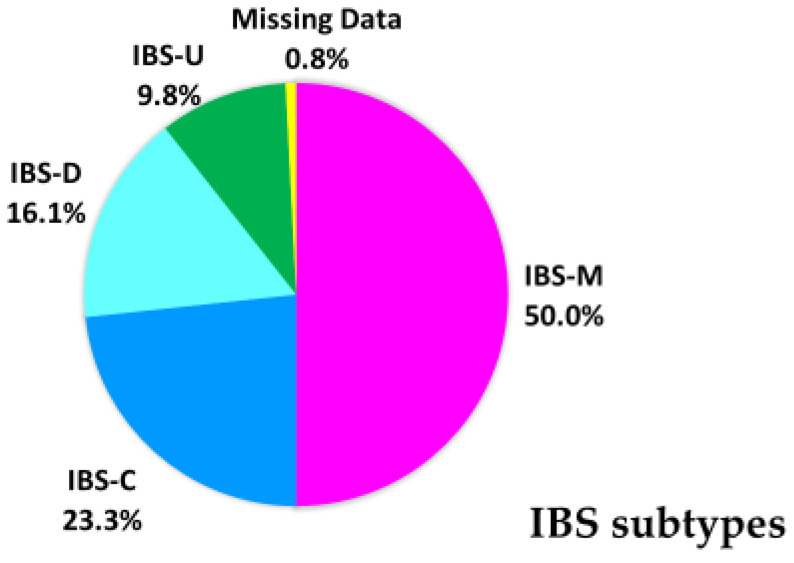
Distribution of IBS subtypes IBS-C: IBS with predominant constipation, IBS-D: IBS with predominant diarrhea, IBS-M: IBS with mixed bowel habits, IBS-U: IBS unclassified.

**Figure 2 ijerph-18-03888-f002:**
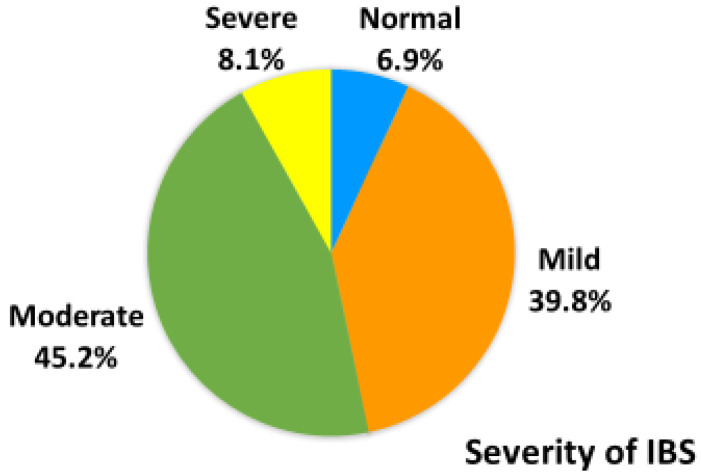
Distribution of severity of IBS female students (N = 193).

**Table 1 ijerph-18-03888-t001:** The distribution of irritable bowel syndrome for female students among three colleges (N = 1894).

College	Department	Number (%)	IBS (%)
College of Medical Technology	Medical Laboratory Science & Biotechnology	136 (7.2)	16 (0.8)
	Biological Science & Technology	49 (2.6)	5 (0.3)
	Optometry	92 (4.9)	9 (0.4)
	Pharmaceutical Science & Technology	13 (0.7)	3 (0.2)
	Cosmetics Application & Management	152 (8.0)	19 (1.0)
	Subtotal	442 (23.4)	52 (2.7)
College of Nursing	Nursing	491 (25.9)	53 (2.8)
	Long-Term Care	81 (4.3)	10 (0.5)
	Health Care Administration	132 (7.0)	11 (0.6)
	Digital Design & Information Management	42 (2.2)	2 (0.1)
	Subtotal	746 (39.4)	76 (4.0)
College of Human Life Science	Occupational Safety & Health	24 (1.3)	3 (0.2)
	Food Nutrition	317 (16.7)	30 (1.6)
	Early Childhood Caring & education	95 (5.0)	8 (0.4)
	Safety Health & Environmental Engineering	18 (0.9)	1 (0.1)
	Hotel & Restaurant Management	199 (10.5)	16 (0.8)
	Sport, Health, & Leisure	53 (2.8)	7 (0.3)
	Subtotal	706 (37.2)	65 (3.4)
	Total	1894 (100)	193 (10.1)

**Table 2 ijerph-18-03888-t002:** Demographic characteristics of female students with or without irritable bowel syndrome (N = 1894).

Variables	Total (N = 1894) *M(SD)/n(%)*	Non-IBS (N = 1701) *M(SD)/n(%)*	IBS (N = 193) *M(SD)/n(%)*	*χ*^2^/*t*	*p* Value
Age	21.59 (1.40)	21.58 (1.43)	21.62 (1.05)	−0.342	0.728
Body height (cm)	159.77 (5.81)	159.76 (5.87)	159.84 (5.24)	−0.189	0.850
Body weight (kg)	54.36 (10.58)	54.42 (10.74)	53.85 (9.10)	0.792	0.429
Body mass index (kg/m^2^)	21.29 (3.80)	21.32 (3.84)	21.08 (3.45)	0.790	0.430
Colleges	1894 (100%)	1701 (100%)	193 (100%)	1.943	0.378
Medical Technology	442 (23.4%)	390 (23.4%)	52 (26.9%)		
Nursing	746 (39.4%)	670 (39.4%)	76 (39.4%)		
Human Life Science	706 (37.2%)	641 (37.2%)	65 (33.7%)		
Years of education	1891 (100%)	1698 (100%)	193 (100%)	0.968	0.616
Sophomore	517 (27.3%)	470 (27.7%)	47 (24.3%)		
Junior	708 (37.5%)	633 (37.3%)	75 (38.9%)		
Senior	666 (35.2%)	595 (35.0%)	71 (36.8%)		
Living status	1879 (100%)	1687 (100%)	192 (100%)	2.519	0.142
at home	788 (41.9%)	717 (42.5%)	71 (37%)		
rental house	1091 (58.1%)	970 (57.5%)	121 (63%)		
Working status	1886 (100%)	1693 (100%)	193 (100%)	0.367	0.545
non-working	1181 (62.6%)	1064 (62.8%)	117 (60.6%)		
working	705 (37.4%)	629 (37.2%)	76 (39.4%)		
Relationship status	1875 (100%)	1683 (100%)	192 (100%)	0.818	0.366
without	1102 (58.8%)	995 (59.1%)	107 (55.7%)		
with	773 (41.2%)	688 (40.9%)	85 (44.3%)		
Internship experience	1890 (100%)	1698 (100%)	192 (100%)	0.011	0.917
without	830 (43.9%)	745 (43.9%)	85 (44.3%)		
with	1060 (56.1%)	953 (56.1%)	107 (55.7%)		
Current internship	1792 (100%)	1608 (100%)	184 (100%)	0.085	0.771
without	1665 (92.9%)	1495 (93.0%)	170 (92.4%)		
with	127 (7.1%)	113 (7.0%)	14 (7.6%)		
Exercise habits	1889 (100%)	1697 (100%)	192 (100%)	0.706	0.401
without (<3 days/week)	676 (35.8%)	602 (35.5%)	74 (38.5%)		
with (≦3 days/week)	1213 (64.2%)	1095 (64.5%)	118 (61.5%)		
Dysmenorrhea	1886 (100%)	1694 (100%)	192 (100%)	61.120	<0.001 **
no	1349 (71.6%)	1258 (74.3%)	91 (47.4%)		
yes	537 (28.4%)	436 (25.7%)	101 (52.6%)		
Food avoidance	1857 (100%)	1670 (100%)	187 (100%)	79.818	<0.001 **
no	1370 (73.8%)	1283 (76.8%)	87 (46.5%)		
yes	487 (26.2%)	387 (23.2%)	100 (53.5%)		
Regular medication (gastrointestinal medication and traditional Chinese medicine)	1876 (100%)	1683 (100%)	193 (100%)	22.360	<0.001 **
no	1581 (84.3%)	1441 (85.6%)	140 (72.5%)		
yes	295 (15.7%)	242 (14.4%)	53 (27.5%)		
Academic performance	1882 (100%)	1690 (100%)	192 (100%)	0.002	0.963
low grade (<80)	938 (49.8%)	842 (49.8%)	96 (50.0%)		
high grade (≧80)	944 (50.2%)	848 (50.2%)	96 (50.0%)		
Class absenteeism	1880 (100%)	1688 (100%)	192 (100%)	22.259	<0.001 **
≦3 days/semester	1333 (71%)	1225 (72.6%)	108 (56.2%)		
>3 days/semester	547 (29%)	463 (27.4%)	84 (43.8%)		

** *p* < 0.001.

**Table 3 ijerph-18-03888-t003:** The level of stress and quality of life between female students with and without irritable bowel syndrome.

Variables	Non-IBS (*n* = *1701*) *M*(*SD*)	IBS (*n* = *193*) *M*(*SD*)	*t*	*p* Value
Stress	26.06 (6.45)	28.87 (6.80)	−5.66	<0.001 **
WHOQOL				
Overall QOL	3.42 (0.72)	3.19 (0.64)	4.73	<0.001 **
Overall Health	3.27 (0.79)	2.80 (0.73)	8.33	<0.001 **
QOL_Physical health	14.46 (2.13)	13.40 (2.31)	6.47	<0.001 **
QOL_Psychological	13.34 (2.56)	12.37 (2.42)	5.04	<0.001 **
QOL_Social Relationships	13.83 (2.32)	13.26 (2.30)	3.19	0.001 *
QOL_Environment	13.67 (2.25)	13.02 (2.27)	3.77	<0.001 **

* *p* < 0.05, ** *p* < 0.001.

**Table 4 ijerph-18-03888-t004:** Logistic Regression for Risk Factors of IBS (N = 1894).

Variables	*β*	S.E.	Wald	OR	*p*	95% C.I.	
						Lower	Upper
Dysmenorrhea (yes/no)	0.925	0.171	29.286	2.522	<0.001 **	1.804	3.526
Class absenteeism (≦3 days/>3 days)	0.571	0.172	11.034	1.771	0.001 *	1.264	2.481
Food avoidance (yes/ no)	1.161	0.171	46.332	3.192	<0.001 **	2.285	4.459
Regular medication (yes/no)	0.383	0.203	3.567	1.467	0.059	0.986	2.183
Stress	0.021	0.016	1.717	1.021	0.190	0.990	1.054
QOL Physical health	−0.140	0.059	5.537	0.870	0.019 *	0.774	0.977
QOL Psycological health	−0.039	0.055	0.490	0.962	0.484	0.863	1.072
QOL Social relationship	0.023	0.052	0.193	1.023	0.660	0.923	1.134
QOL Environment	0.024	0.057	0.183	1.025	0.669	0.916	1.146

* *p* < 0.05, ** *p* < 0.001.

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
