# Peer review of "Exploring Quality of Life, Stress, and Risk Factors Associated with Irritable Bowel Syndrome for Female University Students in Taiwan"

_ijerph, 2021, doi:10.3390/ijerph18083888_

Round 1

Reviewer 1 Report

The manuscript presented by Huan-Hwa and collaborators investigate the prevalence of IBS for female university students and its related factors. The work is interesting but needs improvement in some sections of the results in order to facilitate reading.

  1. Line 39. Approximately 1.1%–45% of world’s population have IBS. That sentence is correct? In conclusions there are other %.
  2. In sections 2.2 and 2.3 a lot of information is repeated
  3. In order to improve reading, I recommend making graphs of section 3.2.
  4. Line 202: What kind of medication did the students take (NSAID?), was information about medication predicted in the survey?. Should be debated in discussion
  5. Line 211. Was the stress categorized in any way, e.g. days per week, do you take measures to combat stress or not?. Should be debated in discussion.
  6. Line 228 what kind of foods avoided the students?. Should be debated in discussion
  7. Line 230. In the West, absenteeism would not be considered a risk factor. I don't know if in Taiwan non-attendance to class brings some kind of consequence to be considered a risk factor. Should be debated in discussion

Author Response

Dear reviewer thank you for all of your kindly suggestions!!

  1. Line 39. Approximately 1.1%–45% of world’s population have IBS. That sentence is correct? In conclusions there are other %.

Ans: 1-1. Yes, the prevalence of IBS approximately 1.1%–45% is correct. According to Lovell & Ford (2012) “Global Prevalence of and Risk Factors for Irritable Bowel Syndrome: A Meta-analysis.” It has demonstrated that prevalence varies strikingly, with varying rates according to diagnostic criteria and geographical area will affect the reported IBS prevalence worldwide.

1-2. In conclusion: The prevalence of IBS in Taiwan is 10.1% in this study (Page 10).

  1. In sections 2.2 and 2.3 a lot of information is repeated.

Ans: We have deleted some sentences in sections 2.2 and 2.3 (Pages 2 , line 79-87).

  1. In order to improve reading, I recommend making graphs of section 3.2.

Ans: We have added two graphs in the section 3.2. (Pages 5-6).

  1. Line202: What kind of medication did the students take (NSAID?), was information about medication predicated in the survey? Should be debated in discussion

Ans: In this study, our participants tend to take gastrointestinal medication and traditional Chinese medicine. In addition, medication is not a predictor of IBS in our study and we have added sentences in discussion (Page 9, line 275-278).

  1. Line 211, Was the stress categorized in any way, e.g. days per week, do you take measures to combat stress or not? Should be debated in discussion.

Ans: In this study, we measured participants’ stress in the past month. Also, we just wanted to know whether stress is a risk factor or not and did not do any intervention measures. We have added sentences in the discussion (Pages 9-10, line 285-304).

  1. Line 228 what kind of foods avoided the students? Should be debated in discussion

Ans: In the discussion section, we have highlighted in red words (Page 9, line 260-266).

  1. Line 230. In the West, absenteeism would not be considered a risk factor. I don't know if in Taiwan non-attendance to class brings some kind of consequence to be considered a risk factor. Should be debated in the discussion.

Ans: In Taiwan students who have physical health conditions, such as dysmenorrhea, IBS, and stomach flu may decide to take a day of menstrual or sick leave. In this study, it is possible that students with IBS missed classes because of the impact of IBS. Higher levels of absenteeism among students suffering from IBS may have cascading effects on their school performance. (Lines 257-260)

Reviewer 2 Report

This is an interesting questionaire-based study regarding IBS in female students of Taiwan. Although somehow outdated, the paper is well-written.

Some recommendations for further improvement: The auhors have to provide the calculations made for the statistical power of the size and prediction of response rate. Secondly, they have to cite or explain, if such translations of questionnaires to Chinese a validated method is indeed in the literature.  To the limitations of the study, it is not only the missing interview, they have to stress that IBS is a diagnosis of exclusion and an extensive diagnostic algorithm is necessitated from endoscopic approach to stool analyses. medication review, breathing tests ultrasound etc , before we label the patient as IBS.  A large amount of patients with IBS do not seek a doctor to perform examinations. These thoughts should be stressed. 

Author Response

Dear reviewer thank you for all of your kindly suggestions!!

  1. The authors have to provide the calculations made for the statistical power of the size and prediction of response rate.

Ans: The sample size was calculated based on G power 3.1.1 version in this study with the effect size of 0.15, a desired significance level of 0.05, two-tailed and a power of 0.95 (Faul, Erdfelder, Buchner, & Lang, 2009). The estimated minimal sample size was 580 participants. Assuming an attrition rate of 30%, a sample size of at least 754 was required (Lines 74-77).

  1. Secondly, they have to cite or explain, if such translations of questionnaires to Chinese a validated method is indeed in the literature.

Ans: In 2.4.2, we did sign the reference [17] (Page 3, line 98).

  1. the limitations of the study, it is not only the missing interview, they have to stress that IBS is a diagnosis of exclusion and an extensive diagnostic algorithm is necessitated from the endoscopic approach to stool analyses. Medication review, breathing tests ultrasound etc, before we label the patient as IBS. A large amount of patients with IBS do not seek a doctor to perform examinations. These thoughts should be stressed.

Ans: We have added sentences in the limitations section (Page10, lines 336-338).
